# Rac Inhibition Causes Impaired GPVI Signalling in Human Platelets through GPVI Shedding and Reduction in PLCγ2 Phosphorylation

**DOI:** 10.3390/ijms23073746

**Published:** 2022-03-29

**Authors:** Raluca A. I. Neagoe, Elizabeth E. Gardiner, David Stegner, Bernhard Nieswandt, Steve P. Watson, Natalie S. Poulter

**Affiliations:** 1Institute of Cardiovascular Sciences, College of Medical and Dental Sciences, University of Birmingham, Birmingham B15 2TT, UK; rxn854@student.bham.ac.uk (R.A.I.N.); s.p.watson@bham.ac.uk (S.P.W.); 2Rudolf Virchow Centre, Institute of Experimental Biomedicine I, University Hospital Würzburg, University of Würzburg, 97080 Würzburg, Germany; stegner@virchow.uni-wuerzburg.de (D.S.); bernhard.nieswandt@virchow.uni-wuerzburg.de (B.N.); 3Division of Genome Science and Cancer, John Curtin School of Medical Research, Australian National University, Canberra, ACT 2601, Australia; elizabeth.gardiner@anu.edu.au; 4Centre of Membrane Proteins and Receptors (COMPARE), Universities of Birmingham and Nottingham, Midlands B15 2TT, UK

**Keywords:** platelets, Rac1, glycoprotein VI, EHT1864, GPVI shedding, phospholipase C gamma 2

## Abstract

Rac1 is a small Rho GTPase that is activated in platelets upon stimulation with various ligands, including collagen and thrombin, which are ligands for the glycoprotein VI (GPVI) receptor and the protease-activated receptors, respectively. Rac1-deficient murine platelets have impaired lamellipodia formation, aggregation, and reduced PLCγ2 activation, but not phosphorylation. The objective of our study is to investigate the role of Rac1 in GPVI-dependent human platelet activation and downstream signalling. Therefore, we used human platelets stimulated using GPVI agonists (collagen and collagen-related peptide) in the presence of the Rac1-specific inhibitor EHT1864 and analysed platelet activation, aggregation, spreading, protein phosphorylation, and GPVI clustering and shedding. We observed that in human platelets, the inhibition of Rac1 by EHT1864 had no significant effect on GPVI clustering on collagen fibres but decreased the ability of platelets to spread or aggregate in response to GPVI agonists. Additionally, in contrast to what was observed in murine Rac1-deficient platelets, EHT1864 enhanced GPVI shedding in platelets and reduced the phosphorylation levels of PLCγ2 following GPVI activation. In conclusion, Rac1 activity is required for both human and murine platelet activation in response to GPVI-ligands, but Rac1’s mode of action differs between the two species.

## 1. Introduction

Platelets are discoid, anucleated blood cells that originate from nucleated precursor cells called megakaryocytes that reside in the bone marrow. The major function is to prevent blood loss upon injury. However, recent studies have shown that they play key roles in inflammation, innate immunity, angiogenesis, and metastasis [1,2,3]. Platelets circulate in the bloodstream in an inactive, resting state, but are activated by collagen exposed in the extracellular matrix after a vascular injury or through atherosclerotic plaque rupture [4]. Platelets’ activation at sites of atherosclerotic plaque erosion/rupture results in arterial thrombosis, which can lead to heart attack or stroke [5]. Platelets express two collagen receptors, glycoprotein (GP) VI, the main signalling collagen receptor, and integrin α2β1 which is involved in the firm adhesion of platelets to collagen [6]. GPVI is an immunoglobulin transmembrane receptor of approximately 65 kDa when glycosylated [7]. At the platelet surface, GPVI is associated with the immunoreceptor tyrosine-based activation motif (ITAM)-containing Fc receptor γ-chain (FcR γ-chain) dimer, which becomes phosphorylated upon collagen binding [8]. This triggers a tyrosine phosphorylation signalling pathway leading to the activation of the spleen tyrosine kinase (Syk), tyrosine phosphorylation of linker for activation of T-cells (LAT) and phospholipase (PL) PLCγ2 among others [9,10]. Once PLCγ2 is activated, it induces Ca^2+^ mobilisation [10] and the activation of protein kinase C (PKC) [11]. Studies using GPVI-deficient murine platelets in ex vivo assays have shown that the receptor is required for platelets to aggregate and secrete in response to GPVI-ligands [12,13]. In vivo, GPVI-depleted or knockout mice only display a slightly impaired haemostasis but are protected in models of arterial thrombosis [14]. Similarly, humans lacking GPVI only present with mild bleeding, but the platelets exhibit defects in aggregation, secretion and phosphatidylserine exposure to collagen. However, adhesion is unaffected [15]. For many years, the conformation of GPVI in resting and activated platelets was debated [16]. Recently, our studies have shown that GPVI is found as a mixture of monomers and dimers in both resting and activated platelets [17], and by using super-resolution microscopy we have shown that GPVI clusters along immobilised collagen fibres in human platelets and at least some of this is dimeric, as it is recognised by a GPVI dimer-specific antibody [18,19]. GPVI also clusters along collagen in mouse platelets [20] but it is unknown whether this is a dimeric GPVI. The intracellular mechanism driving this dimerization and clustering is still unknown, although the observation that inhibiting actin dynamics reduces binding of the dimer-specific antibody suggests a role of the cytoskeleton in dimer formation [18].

GPVI signalling can be regulated at the receptor level by a number of processes. In some situations, GPVI can be internalised [21] but mainly it is cleaved from the membrane by an auto-regulatory process, known as receptor shedding. GPVI shedding occurs after the stimulation of platelets with GPVI ligands (collagen, collagen-related peptide (CRP), convulxin), under high-shear or by other non-physiological stimuli, for instance, by N-ethylmaleimide (NEM), a potent activator of metalloproteinases [22]. Receptor shedding causes irreversible deregulation of the molecular signalling and proteolysis of GPVI into a soluble ectodomain (55 kDa) released into the plasma, and a cytoplasmic domain (10 kDa) that remains anchored in the membrane [23]. Soluble GPVI has a utility as a biomarker to identify platelet activation in pathological settings, such as sepsis progression and mortality [24,25]. In human platelets, GPVI cleavage is mainly mediated by A Disintegrin and Metalloproteinase (ADAM) family proteases with a prominent role of ADAM10 [26]. In murine platelets, GPVI cleavage is mediated by ADAM10 and ADAM17, with evidence that at least one other protease is involved [27]. ADAM10 has been shown to be constitutively active in resting platelets; however, GPVI shedding is minimal. Both ADAM10 activity and shedding increase under high-shear or with NEM treatment [28]. Activation via collagen increases GPVI shedding but not ADAM10 activity, leading to the hypothesis that different stimuli induce shedding via a different mechanism, one of which could be the changes in localisation or conformation of the protease [28,29].

Rho GTPase proteins are found in all eukaryotic cells and cycle between an inactive (GDP-bound) and active (GTP-bound) conformation. They regulate the actin cytoskeleton dynamics in all cell types and, additionally, in platelets, are considered to be primary drivers of platelet signalling and integrin activation [30,31]. One of the most important Rho GTPase in platelets is the protein Ras-related C3 botulinum toxin substrate (Rac). There are three Rac isoforms. Human platelets express Rac1 and, at a lower level, Rac2 [32], whereas, in mouse platelets, only Rac1 is found at significant levels [33]. Rac1 activation occurs upon the binding of collagen to GPVI [34] or integrin α2β1 [35], and in response to thrombin stimulating protease-activated receptors (PARs) [36]. Rac1 plays key roles in lamellipodia formation [33], secretion of dense granules, and platelet aggregation [37]. Of note, in vitro Rac1 activates PLCγ2 [38]. *Rac1^-/-^* murine platelets present with reduced PLCγ2 activity as the mouse line showed impaired inositol trisphosphate (IP3) production and calcium mobilisation. In spite of this, phosphorylation of PLCγ2 in mouse platelets was not affected [33,34]. In other cell types, Rac1 activity has also been shown to play a role in receptor clustering [39,40] and it is not known whether this is the case for platelet GPVI. In order to study the involvement of Rac in human platelet function, pharmacological inhibitors are required. Several Rac inhibitors can block Rac activity in cells. One of the most commonly used is EHT1864, a commercially available inhibitor that completely inhibits the GTP loading of Rac [41]. We have shown previously that a concentration higher than 100 μM EHT1864 causes off-target effects, such as cell apoptosis [42], limiting studies to 50 μM EHT1864 [43,44,45,46], which has been shown to be effective in platelets [47].

In the present study, we investigated the role of Rac1 in GPVI-dependent signalling in human platelets by using EHT1864. We demonstrated that Rac1 inhibition does not disrupt GPVI clustering on collagen fibres. However, we observed that EHT1864 causes an increase in GPVI shedding in both resting and activated human platelets and a decrease in the phosphorylation levels of PLCγ2, suggesting a difference in the signalling between human and mouse platelets.

## 2. Results

### 2.1. EHT1864 Does Not Change GPVI Conformation and Receptor Clustering along Fibrous Collagen

To identify the potential role of Rac1 on GPVI clustering, we used the single-molecule localization microscopy technique dSTORM in combination with the Rac inhibitor (EHT1864). Rac1 inhibition was confirmed (Appendix A) and GPVI organisation was analysed by performing dSTORM imaging (Figure 1(Ai)) of human platelets untreated or pre-treated for 10 min with 30 μM EHT1864 spread on collagen-coated glass. Rac1 inhibition did not affect the alignment of GPVI along fibrous collagen according to a cluster analysis based on DBSCAN (Figure 1(Ai)), where the different GPVI clusters were represented in different colours. The largest clusters were localised along the collagen fibres. The quantitative cluster analysis (Figure 1(Bi–iv)) showed a decreasing trend in the number of detections, cluster density and cluster area on the EHT1864-treated platelets, but this did not reach significance.

We have previously shown with flow cytometry, that GPVI consists of a mixture of monomer and dimers in both resting and CRP-stimulated human platelets [13]. Therefore, we used the same approach to analyse the effect of EHT1864 on GPVI conformation. We labelled GPVI with anti-GPVI 313A10 F (ab’) fragments in two different colours: AF488 (donor fluorophore) and AF546 (acceptor fluorophore). As only one F (ab’) fragment can bind per GPVI molecule, a FRET signal can be observed in GPVI dimers. Thus, the FRET efficiency is a surrogate for the dimer proportion [48]. Of note, the FRET efficiency of resting and CRP stimulated platelets was ~25–30%, independent of Rac1 inhibition (Figure 1C). The FRET assay was controlled by labelling two different and abundant platelet receptors that are known to not dimerise: integrin αIIbβ3, using the antibody MWReg30 (AF488, donor), and GPIX, with anti-GPIX antibody (AF546, acceptor). The FRET efficiency of these receptors was 7% (Figure 1C), which represents the efficiency of only monomers.

These results show that the GPVI clustering along fibrillar collagen and GPVI conformation upon stimulation with the GPVI agonist CRP was not significantly altered by EHT1864 and, therefore, Rac1 does not play a role in these processes.

### 2.2. The Effect of EHT1864 on Human Platelets Activation and GPVI Shedding

As Rac1 did not play a significant role in GPVI organisation and clustering, the effect of Rac1 inhibition on GPVI signalling in human platelets was studied by platelet aggregation experiments with low- and high-dose GPVI agonists (Horm collagen and CRP) (Appendix A). As expected, we observed a dose-dependent effect of EHT1864 on platelet aggregation with 50 μM EHT1864, causing a significant decrease of approximately 50% using either GPVI agonist. Likewise, EHT1864 dose-dependently inhibited platelets spreading on uncoated glass, collagen, and fibrinogen-coated slides without causing a significant change in platelet adhesion to these surfaces (Appendix A). Notably, platelets spreading on collagen in the presence of EHT1864 had a significantly greater percentage (50%) of cells that had adhered but not spread.

Platelet activation was analysed by flow cytometry using the CD41/CD61 antibody PAC-1, which recognises the active form of the integrin. Here, CRP stimulated platelets showed a ~2-fold and ~4-fold reduction in platelet activation when pre-treated for 10 min with 30 μM and 50 μM EHT1864, respectively (Figure 2(Ai)). Granule secretion, assessed by measuring P-selectin expression levels, was also decreased with EHT1864, although this did not reach significance (Figure 2(Aii)). Furthermore, GPVI surface abundance increased after CRP stimulation in control platelets. Moreover, 50 μM EHT1864 prevented this increase and caused a significant reduction of almost 30% in the GPVI exposure at the plasma membrane compared to the controls (Figure 2(Aiii)). When platelets were stimulated with CRP, 50 μM EHT1864 also caused a slight, but not significant, reduction in GPVI levels in resting platelets. A significant trend towards a decreased surface expression in EHT1864-treated CRP-stimulated platelets was also observed for GPV (Figure 2(Aiv)) while the surface abundance of GPIbα (Figure 2(Av)), integrin β3-subunit (Figure 2(Avi)), and CLEC-2 (Figure 2(Avii)) were not affected by EHT1864.

The reduction in GPVI surface abundance could result from receptor shedding or internalisation. Thus, GPVI was investigated by Western blotting in both resting and stimulated platelets using an anti-GPVI-tail antibody [43]. The thiol-modifying agent NEM, a potent inducer of GPVI shedding, was used as a positive control (Figure 2B,C). EHT1864 dose-dependently caused an increase in shedding (as revealed by increasing intensities of the GPVI-tail band), mirroring the reduced GPVI surface expression by flow cytometry. The treatment with the broad matrix metalloproteinase inhibitor GM6001 abrogated EHT1864 induced shedding (Figure 2B,C).

These results suggest that the reduced platelet activation observed in EHT1864-treated platelets is, at least in part, caused by GPVI shedding.

### 2.3. The Effect of EHT1864 on the GPVI Signalling Pathway, Focused on PLCγ2

Next, we investigated tyrosine phosphorylation downstream of GPVI in the presence of vehicle or EHT1864. In mouse platelets, we confirmed the phosphorylation levels of PLCγ2 tyrosine 1216 in resting and CRP stimulated wild type and *Rac1^-/-^* platelets were not different (Figure 3A).

Surprisingly, in human platelets, we observed a decrease of approximately 30% and 40% in p-PLCγ2 levels in platelets pre-treated for 10 min with 30 and 50 μM EHT1864, respectively (Figure 3(Bi,ii)). To assess whether this reduced phosphorylation was the consequence of reduced GPVI surface expression due to EHT1864-induced shedding, we measured PLCγ2 phosphorylation levels in platelets treated with the broad-spectrum metalloproteinase inhibitor GM6001 (Figure 3(Bi,ii)). However, GM6001-treatment did not restore pPLCγ2 levels in EHT1864-treated platelets. The phosphorylation levels of Syk and LAT (Figure 3(Biii,iv)), which are both upstream of PLCγ2, are not affected by EHT1864. This result, combined with the finding that p-Syk remained present along the collagen fibres in both vehicle and EHT1864 treated platelets (Appendix A), supports the literature demonstrating that Rac1 is downstream of Syk and LAT in the signalling pathway [49], and it has a role only in PLCγ2 phosphorylation levels.

## 3. Discussion

Here, we investigated whether the small Rho GTPase Rac1 plays a similar role in human platelets as it does in murine platelets. Previous studies using *Rac1^-/-^* mice have demonstrated that Rac1 is required for lamellipodia formation, aggregate stability [33], and PLCγ2 activation [34]. Rac1 regulates these processes also in human platelets and, in addition, induces GPVI shedding in resting and CRP-stimulated platelets. Interestingly, however, in contrast to mouse platelets, Rac1 modulates the tyrosine phosphorylation of PLCγ2 in human platelets.

Previous studies demonstrated that platelet secretion and aggregation in response to a number of agonists are reduced upon inhibition of Rac1 using the inhibitor NSC23766 [36,50]. This included collagen and atherosclerotic plaque, which is known to activate platelets through GPVI. They also observed a defect in thrombin receptor activating peptide (TRAP)-induced platelet aggregation, which differs from that seen in mice [50]. We capitalised on the Rac1 inhibitor EHT1864, which is more specific, at least up to a maximum concentration of 50 μM [42]. We observed that EHT1864 partially blocked the aggregation of platelets stimulated with GPVI agonist (CRP and Horm collagen) and reduced platelet activation markers, in line with findings from previous studies.

Rac1 also mediates the reorganisation of the actin cytoskeleton during platelet activation [37]. We observed that EHT1864 significantly affected the lamellipodia formation and therefore the degree of spreading, confirming previous data from *Rac1^-/-^* mice [33]. However, in line with the literature [51], the inhibition of Rac1 did not change the ability of platelets to adhere to different surfaces. In other cells, the mediation of Rac1 in the actin cytoskeleton regulates the receptor dynamic [39,52]. Hence, our hypothesis for this work was that Rac1 could also play a role in GPVI clustering and conformation. We used dSTORM, a single-molecule super-resolution microscopy method, to analyse GPVI clusters along collagen fibres. Our DBSCAN cluster plots suggest that EHT1864 does not disrupt GPVI clustering on human platelets. This complements another study conducted by our group, where we looked at the effect of Syk and Src inhibitors on the GPVI clustering, reporting that the inhibition of these two important kinases does not disturb the clustering either [19]. These data suggest that the binding of GPVI to collagen fibres induces GPVI clustering independently of the downstream signalling pathway. As a follow-up to our last study, where FRET was used to demonstrate that GPVI is found in a mixture of monomers and dimers in resting and CRP-activated platelets [17], we studied the effect of Rac inhibition on GPVI dimerisation by FRET. EHT1864 does not change the percentage of dimers present in resting or activated platelets.

Previously, we conducted a study [53] where we verified that EHT1864 concentrations up to 200 μM EHT1864 do not cause a significant alteration in the surface abundance of platelet receptors in *Rac1^-/-^* platelets. In the present study, in human platelets, we showed that 50 μM EHT1864 significantly reduced GPVI and GPV exposure levels in CRP-stimulated platelets. However, the exposure levels of other main platelet receptors, namely GPIbα, integrin β3-subunit, and CLEC-2 were not affected by EHT1864. Furthermore, it is well known that GPVI and GPV shedding occurs on activated platelets through the metalloproteinase ADAM10 [26,54], whereas GPIbα is shed only by ADAM17 [54] and CLEC-2 shedding is independent of ADAM10 [55]. All of these data suggest that the EHT1864-induced reduction in GPVI and GPV exposure is a mechanism in which ADAM10 is involved.

This study presents the novel finding that Rac1 also plays a role in GPVI shedding, where inhibition with EHT1864 causes a significant increase in GPVI cleavage on both resting and CRP-activated platelets. As inhibition of metalloproteinases prevented GPVI shedding, ADAM10 might be the underlying protease. While the present study is the first to describe the role of Rac1 in GPVI shedding, this correlates with data from other cell lines where Rac1 also plays a role in membrane receptor CD44 shedding through ADAM10 [56]. Notably, there is a link between calmodulin (CaM) and Rac1. CaM binds Rac1 in platelets and affects Rac1 activation as the treatment with the CaM inhibitor N-(6-aminohexyl)-5-chloro-1-naphthalenesul-fonamide (W7), also causes the inhibition of Rac1 [57]. Of note, W7-treatment of platelets results in GPVI shedding [58]. As W7 does not enhance ADAM10 activity, the current concept is that the dissociation of CaM from GPVI enables ADAM10 to easily reach the cleavage site of GPVI, thereby promoting shedding [28]. Our study adds Rac1 into this picture, indicating that Rac1 activity might be required to keep CaM associated with GPVI. Alternatively, Rac1 could play its role by mediating tetraspanins (Tspan), which are known to modulate ADAM10 activity [59]. In platelets, Tspan15 interacts with ADAM10 and this molecular scissor complex is involved in GPVI shedding [60]. Recently, it has been shown that Rac1 binds the cytosolic domain of Tspan15 [61], although Rac1’s role is not yet clear. This interaction could be an additional factor in modulating GPVI surface expression via Rac1. Clearly, further research is needed to reveal the exact mechanisms underlying the regulation of GPVI shedding.

We have shown that EHT1864 causes a significant decrease in PLCγ2 phosphorylation levels and that this is independent of EHT1864-induced GPVI shedding. Pleines et al. [34] observed that there was reduced IP3 production, but unaltered PLCγ2 phosphorylation in *Rac1^-/-^* platelets. In this study, we have confirmed, using the same Rac1 deficient mouse line under the same stimulation conditions used for human platelets, that the PLCγ2 phosphorylation level does not vary between WT and Rac1 deficient platelets [27]. We hypothesise that this difference is innate to each species, and most likely in human platelets mainly, Rac1 is in control of PLCγ2 phosphorylation, while in mouse, as for the GPVI shedding, there are other proteins playing a crucial role. Further work needs to be conducted to unravel the exact mechanism that controls PLCγ2 phosphorylation in murine platelets.

In conclusion, as in murine platelets, Rac1 is also crucial for human platelet aggregation and lamellipodia formation and plays an additional role in GPVI shedding and PLCγ2 phosphorylation. Despite the role of Rac1 in remodelling the actin cytoskeleton, it does not induce any significant disruption to the GPVI clustering and dimerisation. These results highlight the species differences in Rac1’s role in the GPVI signalling pathway.

## 4. Materials and Methods

### 4.1. Antibodies and Reagents

The following antibodies were used: for flow cytometry, PE-conjugated anti-human GPVI (clone: HY101; BD Biosciences, San Diego, CA, USA); for FRET, the F (ab’) fragment of the anti-human GPVI (clone: 313A10) [17]; for super-resolution microscopy, the 1G5 F (ab’) fragment [62] and for GPVI shedding analysis, the affinity-purified anti-GPVI cytoplasmic tail IgG [63] (both gifts from Dr E. Gardiner, Australia) were used. Anti-GPIX (clone: p0p6) [64], anti-integrin αIIb (clone: MWReg30) [65], anti-integrin β3-subunit (clone: EDL1) [66], anti-GPV (clone: 10C10) and anti-CLEC-2 (clone: HEL1) [67] antibodies were all purified and fluorophore-conjugated in house. Anti-PLCγ2 pY1217 antibody was purchased from Santa Cruz Biotechnology (Santa Cruz, CA, USA); anti-SYK Py525/526, anti-GAPDH and anti-LAT antibodies were from Cell Signalling (Danvers, MA, USA); anti-LAT pY200 was from Abcam (Cambridge, UK); Alexa Fluor^®^ 488-Phalloidin was from Invitrogen (Carlsbad, CA, USA); Alexa Fluor^®^ 647 labelled anti-human CD41/CD61 antibody (clone: PAC-1) was from BioLegend (San Diego, CA, USA); APC-labelled anti-human CD42b (clone: HIP1) and FITC-labelled anti-human P-selectin (clone: AK-4) antibodies were both from BD Biosciences (San Diego, CA, USA).

The Rac1 inhibitor EHT1864 was purchased from Tocris Bioscience (Bristol, UK), and the ADAMs inhibitor GM60001 was purchased from Sigma Aldrich (Burlington, MA, USA).

### 4.2. Animals

The generation of the mice with *Rac1^-/-^* platelets was described in previous studies from our group [34]. The animal experiments were performed following the regulations and guidelines approved by the local authorities (District Government of Lower Franconia).

### 4.3. Human Platelet Preparation

Washed human platelets were obtained as described previously [68] from healthy donors. Platelets were diluted in Tyrode’s-HEPES buffer (129 mM NaCl, 0.34 mM Na_2_HPO_4_, 2.9 mM KCl, 12 mM NaHCO_3_, 20 mM HEPES, 5 mM glucose, 1 mM MgCl_2_; pH 7.3) to the desire concentration: for aggregation, 2 × 10^8^/mL; for spreading and flow cytometry assays, 2 × 10^7^/mL; for Western blotting, 5 × 10^8^/mL.

Washed murine platelets were prepared as described previously [69]. The platelets were diluted to 5 × 10^8^/mL for the Western blotting assay.

### 4.4. Flow Cytometry Assay

Human washed platelets were pre-incubated with vehicle or EHT1864 (30 or 50 μM) for 10 min at 37 °C. Then, platelets were stimulated with vehicle or CRP (5 μg/mL) and incubated with APC-conjugated anti-human CD41/CD61 antibody (clone: PAC-1), FITC-conjugated anti-human P-selectin, PE-conjugated anti-human GPVI (clone: HY101), FITC-conjugated anti-human GPV, APC-conjugated anti-human GPIbα, FITC-conjugated anti-human integrin β3 and FITC-conjugated anti-human CLEC-2 antibodies for 10 min at 37 °C and 10 min at room temperature (RT).

Samples were measured with a CytoFLEX flow cytometer (Beckman Coulter, Brea, CA, USA) and the analysis was performed using the acquisition software CytExpert (Beckman Coulter, Brea, CA, USA).

### 4.5. GPVI Shedding Assay and Protein Phosphorylation

For the GPVI shedding and protein phosphorylation assays, human washed platelets were incubated with eptifibatide (9 μM) in the presence or absence of the broad matrix metalloproteinase inhibitor, GM6001 (250 μM), for 10 min at 37 °C. Then, platelets were treated with different concentrations of EHT1864 (0, 30, and 50 μM) for 10 min at 37 °C. Later, platelets were stimulated with the GPVI agonist CRP (5 μg/mL) for 20 min at 37 °C under shacking conditions (900 rpm). Activation was stopped, and protein lysate was obtained by the addition of 5 × reducing sodium dodecyl sulphate (SDS) sample buffer (10 mg/mL SDS, 25% 2-mercaptoethanol, 50% glycerol, 25% stacking buffer, and a small amount of Brilliant Blue) supplemented with 500 mM of dithiothreitol (DTT) for 15 min on ice. Protein lysates were separated via SDS-PAGE on 4–12% Bis-Tris NuPage gels (Invitrogen, Carlsbad, CA, USA). Western blotting was performed as described previously [70]. The polyvinylidene difluoride membrane was incubated with the anti-GPVI cytoplasmic tail IgG (1 μg/mL), for protein phosphorylation with anti-PLCγ2 Y1217 (1:250), Syk Y525/526 (1:500), LAT Y200 (1:500), and for loading control, with anti-LAT (1:500) and GAPDH (1:1000) antibodies diluted in blocking buffer (4% bovine serum albumin (BSA) diluted in 1 × tris-buffered saline, 0.1% Tween 20) overnight at 4 °C. The following day, membranes were incubated for 1 h at RT with an anti-rabbit IgG-HRP conjugated antibody (1:10,000) (GE Healthcare, Little Chalfont, UK) as a secondary antibody. Membranes were developed using ECL-Super Signal West Pic Plus chemiluminiscence substrate (ThermoFisher Scientific, Waltham, MA, USA) and imaged using the Odyssey imaging system (LI-COR Biosciences, Lincoln, NE, USA). The quantification of the band intensity was performed using Image Studio Lite v5.2 software. The results were transferred to Excel (Microsoft, Redmond, WA, USA) and the relative values were normalised to the loading controls, GAPDH or pan-LAT. Next, the values were also normalised to the vehicle control CRP-stimulated sample. The percentage of GPVI shedding was calculated as previously described [19] using the band intensities as follows:(1)$$\%\;\mathrm{GPVI}\;\mathrm{shedding}=\frac{\mathrm{GPVI}\;\mathrm{tail}}{\left(\mathrm{GPVI}\;\mathrm{full}\;\mathrm{length}+\mathrm{GPVI}\;\mathrm{Tail}\right)}$$

### 4.6. Direct Stochastic Optical Reconstruction Microscopy (dSTORM)

Washed platelets were incubated for 10 min at 37 °C with EHT1684 (30 μM) or vehicle control prior to spreading on collagen-coated (10 μg/mL), 5 mg/mL BSA blocked, glass-bottomed MatTek dishes (MatTek Corporation, Ashland, MA, USA) for 45 min. Platelets were fixed in formalin for 10 min, permeabilised with 0.1% (*v*:*v*) Triton X-100 in PBS for 5 min and blocked in 1% (*w*/*v*) BSA supplemented with 2% (*v*:*v*) goat serum. Then, platelets were labelled with anti-GPVI 1G5 F (ab’) fragment (2 μg/mL) for 1 h at RT. Phalloidin-Alexa Fluor^®^ 488 conjugated (1:500) and anti-mouse IgG-Alexa Fluor^®^ 647 conjugated (1:300) antibodies were diluted in blocking buffer and incubated for 1 h at RT. Labelled platelets were visualised in dSTORM blinking buffer (100 mM MEA, 50 μg/mL glucose oxidase, and 1 μg/mL catalase diluted in PBS, pH 7.5) that allows the photoswitching performance. The images were obtained using the 100 × objective (NA 1.47) in the TIRFM mode of a NIKON N-STROM microscope. dSTORM super-resolved images were obtained as previously described [18].

### 4.7. Cluster Analysis

The reconstruction of the dSTORM images was performed using the ThunderSTORM ImageJ plugin [71]. Then, the cluster analysis was performed as described in previous studies from our group using a density-based spatial clustering of applications with the noise (DBSCAN) clustering analysis tool [70], setting the radius of the local neighbourhood to 50 nm. This allowed the quantification of the number of detections, number of clusters, cluster density and cluster area of each sample. The number of detection and number of clusters was calculated per unit area. The surface area of the platelets in the field of view was calculated using the ImageJ software, drawing a line around each platelet.

### 4.8. Förster Resonance Energy Transfer (FRET) in Platelets

Flow cytometric FRET analysis was performed as previously described [17]. Human washed platelets were pre-incubated with 30 μM EHT1864 for 10 min, then labelled with 50 μg/mL of human anti-GPVI F (ab’) fragment 313A10 conjugated with Alexa Fluor^®^ 488 (donor), 546 (acceptor) or unlabelled. As a control for the assay anti-integrin, αIIbβ3 (clone: MWReg30) and anti-GPIX (clone: p0p6) antibodies conjugated with Alexa Fluor^®^ 488 and 546, respectively, were used in murine washed platelets. Immediately after, platelets were stimulated with vehicle (PBS) or CRP (10 μg/mL). For the measurements, the FACSAria flow cytometer (BD Biosciences, San Diego, CA, USA) was used. All measurements were performed using the FACSDiva software (BD Biosciences); 50,000 platelets were analysed per sample. During FRET, it was assumed that there was an energy transfer from an excited donor fluorophore (488) to an acceptor fluorophore (546). The Fret efficiency (E) was calculated according to the following equation [72]:(2)$$E=1\;-\;\frac{I_{\mathrm{Donor}+\mathrm{Aceptor}\;}{-\;I}_{\mathrm{Background}}}{I_{\mathrm{Donor}\;}{-\;I}_{\mathrm{Background}}}$$

When there was FRET, a reduction in the mean fluorescence intensity (I) of the donor in the presence of the acceptor (I_Donor+Aceptor_) compared to the mean fluorescence intensity of the donor alone (I_Donor_) occurred.

### 4.9. Statistical Analysis

Data analysis was performed using Prism v8.0.2 (GraphPad Software, La Jolla, CA, USA). Each experiment was performed at least three times and the results are shown as mean ± standard deviation (SD) unless otherwise stated. The statistical tests used for each experiment are detailed in the figure legends. A *p*-value < 0.05 was considered significant.

## Figures and Tables

**Figure 1 ijms-23-03746-f001:**
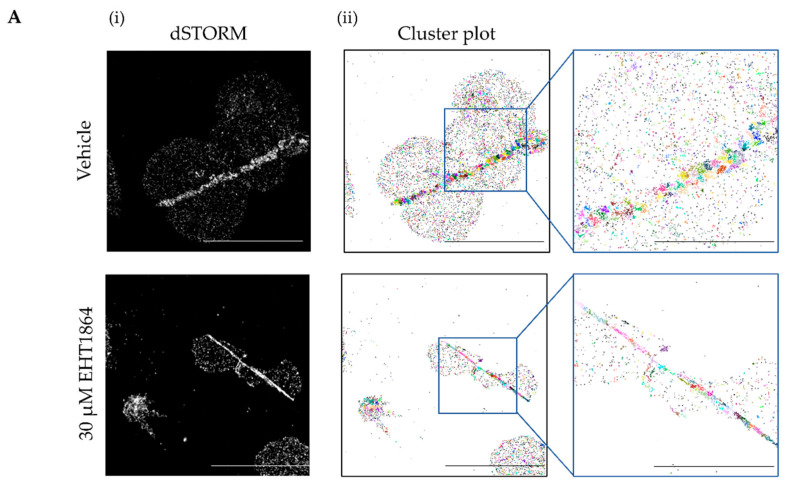
EHT1864 does not change GPVI receptor clustering and dimerisation on collagen. (**A**) Untreated and EHT1864-treated human platelets labelled with 1G5-(Fab’) fragment were imaged using the NIKON N-STORM microscope and three different fields of view were captured per condition. (**Ai**) Representative dSTORM reconstructed images. The reconstruction of 20,000 frames per image was completed using the ImageJ ThunderSTORM plugin. Scale bar: 10 μm. (**Aii**) Representative GPVI cluster plot was obtained using KNIME software with an algorithm based on density-based spatial clustering of applications with noise (DBSCAN). Scale bar: 10 μm (left cluster plot) and 5 μm (right magnified cluster plot). (**B**) Quantification of the effect of EHT1864 (30 μM) on the GPVI cluster analysis. Relative quantification of (**Bi**) number of detections, (**Bii**) number of clusters, (**Biii**) number of detections per cluster, and (**Biv**) cluster area (in nm^2^). (*n* = 3) Mean ± SEM, by paired t-test. (**C**) FRET efficiency of GPVI on resting or CRP (10 μg/mL) stimulated platelets, untreated or treated with EHT1864 (30 μM) for 10 min at 37 °C, labelled with anti-human GPVI (313A10) F (ab’) fragments and as a negative control, anti-integrin αIIbβ3 and anti-GPIX antibodies (*n* = 3). Mean ± SEM, by paired *t*-test.

**Figure 2 ijms-23-03746-f002:**
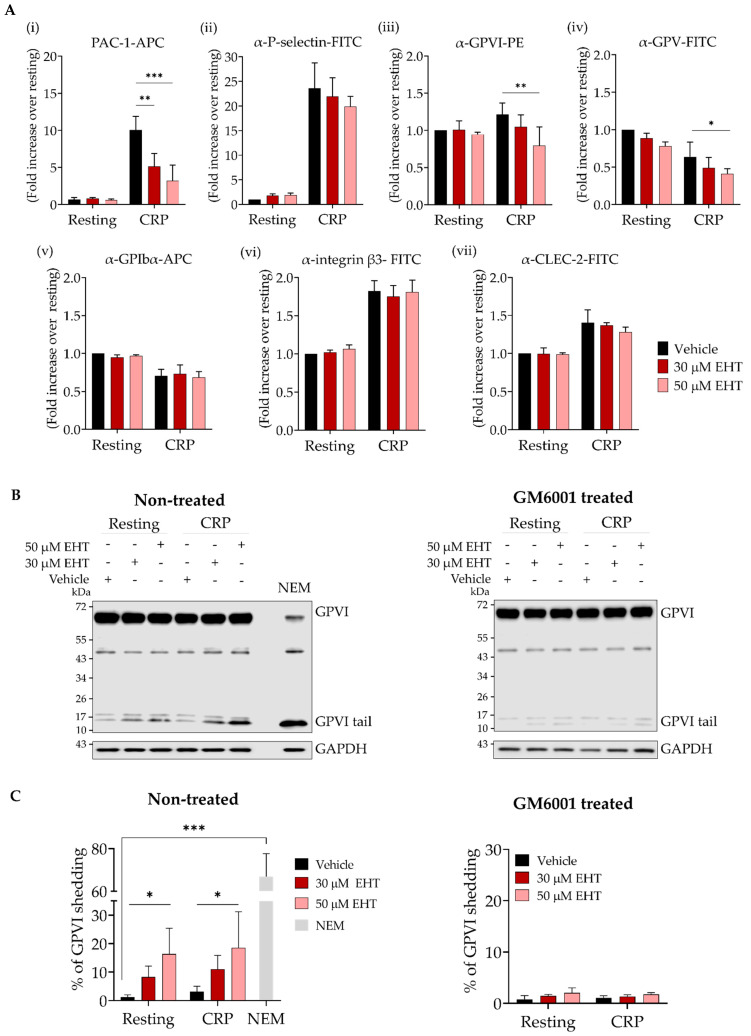
The effect of EHT1864 on GPVI-induced platelet activation and GPVI shedding. (**A**) Dose-dependent reduction in platelet activation and GP exposure by EHT1864. (**Ai**) Flow cytometric analysis of resting or CRP-activated (5 μg/mL) human platelet activation, pre-treated with vehicle (PBS) or EHT1864 (30 or 50 μM) for 10 min at 37 °C, and labelled with Alexa Fluor^®^ 647 conjugated anti-human activated CD41/CD61 (PAC-1) antibody, (**Aii**) FITC-labelled anti-P-selectin antibody, (**Aiii**) PE-conjugated anti-human GPVI (HY101) antibody, and FITC or APC-conjugated anti-human antibodies for (**Aiv**) GPV, (**Av**) GPIbα, (**Avi**) integrin β3-subunit and (**Avii**) CLEC-2 (*n* = 3). (**B**) GPVI shedding induced by EHT1864. Representative Western blots showing GPVI shedding on non-treated and GM6001 (250 μM, broad matrix metalloproteinase inhibitor) treated human platelets, non-stimulated or stimulated with CRP (5 μg/mL) and pre-incubated with vehicle (PBS) or EHT1864 (30 or 50 μM) or stimulated with NEM (2 mM). (**C**) Quantification of the percentage GPVI shedding induced by EHT1864 on non-treated or GM6001 (250 μM) treated human platelets (*n* = 3). Mean ± SD * *p* < 0.05, ** *p* < 0.01, *** *p* < 0.001 by two-way ANOVA (comparing vehicle vs EHT1864 treated platelets). Shedding induced by NEM compared to vehicle resting platelets.

**Figure 3 ijms-23-03746-f003:**
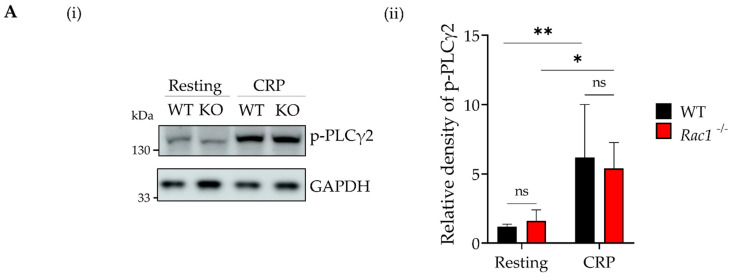
The effect of Rac1 inhibition on the GPVI signalling pathway in murine and human platelets. (**A**) Determination of protein phosphorylation on Rac1-deficient platelets. (**Ai**) Representative image of Western blotting of p-PLCγ2 on non-stimulated and CRP (5 μg/mL) stimulated platelets. (**Aii**) Quantification of the band intensity of p-PLCγ2. (**B**) Decrease in human platelets phosphorylation levels induced by EHT1864. (**Bi**) Representative images of Western blotting of GPVI signalling phosphorylation, including PLCγ2, Syk, and LAT in non-treated or GM6001 (250 μM, broad matrix metalloproteinase inhibitor) treated human platelets, unstimulated or stimulated with CRP (5 μg/mL) and pre-incubated with vehicle (PBS) or EHT1864 (30 or 50 μM) for 10 min at 37 °C. Quantification of the band intensity of (**Bii**) p-PLCγ2, (**Biii**) p-Syk and (**Biv**) p-LAT phosphorylation level induced by EHT1864 in non-treated or GM6001 (250 μM) treated human platelets (*n* = 3). Mean ± SD * *p* < 0.05, ** *p* < 0.01 and **** *p* < 0.0001 by two-way ANOVA (comparing vehicle vs. EHT1864 treated platelets).

## Data Availability

All data, except the raw dSTORM data, are included in this manuscript. Raw microscopy data is available on request from rxn854@student.bham.ac.uk.

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
