# Peer review of "Rac Inhibition Causes Impaired GPVI Signalling in Human Platelets through GPVI Shedding and Reduction in PLCγ2 Phosphorylation"

_ijms, 2022, doi:10.3390/ijms23073746_

Round 1

Reviewer 1 Report

The article ‘Rac Inhibition Causes Impaired GPVI Signaling in Human Platelets Through GPVI Shedding and Reduction of PLCγ2 Phosphorylation' by Neagoe et al. discusses the effects of Rac1 inhibition on human platelets. The authors have used Rac1-specific inhibitor EHT1864 and dSTORM, FRET, among other techniques and assays to study the human platelet activation, aggregation, spreading, PLCγ2 phosphorylation, and GPVI shedding. The authors argue that inhibiting Rac1 in human platelets affects GPVI shedding but not clustering, phosphorylation of PLCγ2, platelet aggregation, and spreading. The study is informative and relevant for the readership of 'IJMS.' The study design and experimental approaches are appropriate. However, the authors need to address some critical concerns in the current version of the manuscript before its publication. The major and minor issues are listed below.

Major: 

GPVI cluster density can depend on the substrate used. Have the authors considered using collagen III as substrate, which affects cluster density, as reported in Poulter et al., J Thromb Haemost, 2017? 

Have the authors seen changes in GPVI clusters localization within the areas of phosphotyrosine upon Rac inhibition? 

Can the authors comment on whether activating Rac in platelets specifically increases p-PLCγ2 and not Syk and LAT? 

How does affecting Rac change the signaling activity in platelets? For example, how does downstream calcium or granule secretion are affected? 

Have the authors attempted to quantify soluble ectodomain GPVI fraction? 

Clustering of GPVI-dimers is proposed as one of the mechanisms facilitating platelet activation and signaling persistence. Can the authors comment on the alternate hypothesis of Rac dependent platelet activation in the absence of changes in clustering?  

The authors should include images showing the effects of Rac1 on platelet aggregation. 

How do the authors explain the effects of EHT1864 affecting spreading but not adhesion? 

Minor: 

It would be helpful to include CRP-induced platelet aggregation data of light transmission aggregometry at CRP (5 μg/ml) as this concentration was used in experiments. 

Discussion:

This complements another study conducted by our group where we looked at the effect of Syk and Src inhibitors on the GPVI clustering, reporting that the inhibition of these two important kinases do not disturb the clustering neither [15]. These dat suggest that binding of GPVI to collagen fibers induces GPVI clustering independently of downstream signalling pathway.

A better word choice would be ‘either’ in the first sentence.’ There is also a typographical error in the second sentence. 

EHT1864 does not cause an alteration on the percentage of dimers in neither resting nor activated Platelets

A period sign is missing.  

Reviewer 2 Report

The aim of this manuscript is to analyze the role of Rac1 in GPVI-dependent signalling, in human platelets, through the use of EHT1864, a specific Rac1 inhibitor. In this context, the authors evaluate platelet activation, aggregation, protein phosphorylation and GPVI clustering.

As regards the main topic, it is interesting and certainly of great scientific and clinical impact: in fact, this manuscript touches a significant area. Overall, the contents are rich and the authors also give their deep insight for some works. Even if the manuscript provides an organic overview, with a densely organized structure and based on well-synthetized evidence, there are aspects to be mentioned, to make the article fully readable. For these reasons, the manuscript requires minor changes.

Please find below an enumerated list of comments on my review of the manuscript:

INTRODUCTION:

Platelets represent a cellular subgroup of the elements, circulating in the bloodstream, which exert a pivotal role in responding to vessel injuries, regulating angiogenesis and the innate immunity, as reported by several and recent studies (see, for reference: Bianchi, S.; Torge, D.; Rinaldi, F.; Piattelli, M.; Bernardi, S.; Varvara, G. Platelets’ Role in Dentistry: From Oral Pathology to Regenerative Potential. Biomedicines 2022, 10, 218. https://doi.org/10.3390/biomedicines10020218). The minor point of this paper is associated to the reference: the manuscript will benefit from providing an organic and, at the same time, accessible introduction to platelets, providing sufficient information for the non-expert while also achieving a balance of detail for those with more expertise in the field. This is the additional point, which makes this manuscript original, in comparison to published literature.

As regards the pivotal role of Rho GTPase protein Rac, due to its dynamic contribute to cytoskeletal reorganization, platelet activation and integrin activation, Rac is considered a primary driving for platelets’ signalling, as suggested by recent studies (see, for reference: Comer SP. Turning Platelets Off and On: Role of RhoGAPs and RhoGEFs in Platelet Activity. Front Cardiovasc Med. 2022 Jan 6;8:820945. doi: 10.3389/fcvm.2021.820945. PMID: 35071371; PMCID: PMC8770426).

As regards the section of methods, there is a specific and detalied explanation for the majority of methods used in this study: this is particularly significant, since the manuscript relies on a multitude of methodological and statistical analysis, to derive its conclusions. The methodology applied is overall correct, the results are reliable and adequately discussed. At the same time, this manuscript show rich content, providing a deep insight for some works: I found it to be well-written.

Also the discussion is well organized and densely presented, based on well-synthetized data, while the conclusion of this manuscript is perfectly in line with the main purpouse of the paper: they are well written and present an adequate balance between the description of previous findings and the results presented by the authors.

In conclusion, this manuscript is densely presented and well organized, based on well-synthetized evidences. The authors were lucid in their style of writing, making it easy to read and understand the message, portrayed in the manuscript. Besides, the methodology design was rigorous and appropriately implemented within the study. However, many of the topics are very concisely covered. This manuscript provided a comprehensive review of current knowledge in this field. Moreover, this research have futuristic importance and could be potential for future research. However, the minor concern of this manuscript is with the introductive section: for these reasons, I have minor comments only for the introductive section, for improvement before acceptance for publication. The article is accurate and provides relevant information on the topic and I suggest minor changes to be made in order to maximize its scientific impact. I would accept this manuscript, if the comments are addressed properly.

Round 2

Reviewer 1 Report

After careful examination of the response of the authors, and the changes made in the manuscript, I gather that the revised version of the manuscript has addressed the major concerns raised in the previous version of the paper. I am therefore happy to endorse the publication of this paper in the journal.